# Methodologies and MR Parameters in Quantitative Magnetic Resonance Neurography: A Scoping Review Protocol

**DOI:** 10.3390/mps5030039

**Published:** 2022-05-06

**Authors:** Fabian Balsiger, Benedikt Wagner, Johann M. E. Jende, Benjamin Marty, Martin Bendszus, Olivier Scheidegger, Felix T. Kurz

**Affiliations:** 1Support Center for Advanced Neuroimaging (SCAN), Institute for Diagnostic and Interventional Neuroradiology, Inselspital, Bern University Hospital, University of Bern, 3010 Bern, Switzerland; benedikt.wagner@insel.ch (B.W.); olivier.scheidegger@insel.ch (O.S.); 2Department of Neurology, Inselspital, Bern University Hospital, University of Bern, 3010 Bern, Switzerland; 3Department of Neuroradiology, Heidelberg University Hospital, 69120 Heidelberg, Germany; johann.jende@med.uni-heidelberg.de (J.M.E.J.); martin.bendszus@med.uni-heidelberg.de (M.B.); 4NMR Laboratory, Neuromuscular Investigation Center, Institute of Myology, 78013 Paris, France; b.marty@institut-myologie.org; 5NMR Laboratory, CEA, DRF, IBFJ, MIRCen, 78013 Paris, France; 6Division of Radiology, German Cancer Research Center, 69120 Heidelberg, Germany

**Keywords:** magnetic resonance imaging, magnetic resonance neurography, peripheral nerve, quantitative, scoping review

## Abstract

Magnetic resonance neurography (MRN), the MR imaging of peripheral nerves, is clinically used for assessing and monitoring peripheral neuropathies based on qualitative, weighted MR imaging. Recently, quantitative MRN has been increasingly reported with various MR parameters as potential biomarkers. An evidence synthesis mapping the available methodologies and normative values of quantitative MRN of human peripheral nerves, independent of the anatomical location and type of neuropathy, is currently unavailable and would likely benefit this young field of research. Therefore, the proposed scoping review will include peer-reviewed literature describing methodologies and normative values of quantitative MRN of human peripheral nerves. The literature search will include the databases MEDLINE (PubMed), EMBASE (Ovid), Web of Science, and Scopus. At least two independent reviewers will screen the titles and abstracts against the inclusion criteria. Potential studies will then be screened in full against the inclusion criteria by two or more independent reviewers. From all eligible studies, data will be extracted by two or more independent reviewers and presented in a diagrammatic or tabular form, separated by MR parameter and accompanied by a narrative summary. The reporting will follow the guidelines of the Preferred Reporting Items for Systematic reviews and Meta-Analyses extension for Scoping Reviews (PRISMA-ScR). Upon completion, the scoping review will provide a map of the available literature, identify possible gaps, and inform on possible future research. SCOPING REVIEW REGISTRATION: Open Science Framework 9P3ZM.

## 1. Introduction

Magnetic resonance neurography (MRN) [1,2], the MR imaging of peripheral nerves, is gaining increasing attention for assessing and monitoring peripheral neuropathies. Several clinical studies have highlighted the potential of MRN, and MRN is now clinically used as a complementary diagnostic tool for gold standard neurological examination and electrodiagnostic studies [3,4,5,6,7,8,9,10,11,12,13,14,15]. In clinical practice, MRN is often employed qualitatively, i.e., weighted MR imaging relying on the subjective evaluation by neuroradiologists. However, quantitative MRN for an objective, repeatable, and reproducible evaluation of peripheral neuropathies in clinical practice would be advantageous.

Quantitative MRN is nowadays actively being investigated for the extraction of MR biomarkers of peripheral neuropathies. Several non-systematic reviews [3,4,5,6,7,8,9,10,11,12,13,14,15] summarized the potential of MRN in general and coverd advances in quantitative MRN to some extent. Particularly, Chen et al. [12] provided a summary of the possible biomarkers ranging from pure MR parameters like T2 relaxation, diffusion tensor parameters, and magnetization transfer to indirect MR parameters like cross-sectional area extracted from weighted imaging. To identify conducted and ongoing systematic reviews or scoping reviews on quantitative MRN, a preliminary search of MEDLINE (PubMed), the Cochrane Database of Systematic Reviews, PROSPERO, and JBI Evidence Synthesis was conducted on 18 March 2022. Four conducted systematic reviews were identified. Wade et al. [16] summarized diffusion tensor parameters of the brachial plexus of nine studies in a systematic review identifying normative values. Schreiber et al. [17] systematically reviewed ultrasound and MRN in amyotrophic lateral sclerosis, which also included studies reporting quantitative MRN. Van der Cruyssen et al. [18] identified studies on trigeminal neuropathy, including some that also report quantitative MRN. Evans et al. [19] reviewed MRN biomarkers in diabetic and HIV-associated peripheral neuropathies. Therefore, to the best of our knowledge, there exists no rigorous evidence synthesis mapping the available methodologies and normative values of quantitative MRN of human peripheral nerves, independent of the anatomical location and type of neuropathy, in the literature.

We propose a scoping review to provide an evidence synthesis in quantitative MRN. The scoping review will consolidate the literature on the emerging field of quantitative MRN of human peripheral nerves. Both the characteristics and ranges of the methodologies and the measured values of MR parameters will be extracted and reported. Therefore, it will provide a map of the available literature, identify possible gaps, and inform on possible future research. According to the best practices for the conduct of scoping reviews [20,21,22], we established the protocol at hand describing the methodology of the proposed scoping review.

### Review Question

The review question of the scoping review will be: What are the characteristics and ranges of methodologies and measured values of MR parameters in quantitative magnetic resonance imaging of human peripheral nerves in vivo?

## 2. Methods

A systematic literature search, literature selection, and literature synthesis will be used to answer the review question through a scoping review. The scoping review will be conducted in accordance with the JBI (Joanna Briggs Institute) methodology for scoping reviews [21,23]. The reporting of the scoping review will be in accordance with the Preferred Reporting Items for Systematic reviews and Meta-Analyses extension for Scoping Reviews (PRISMA-ScR) [20]. We registered the scoping review in the Open Science Framework (https://osf.io/9p3zm, accessed on 28 April 2022). The Preferred Reporting Items for Systematic reviews and Meta-Analyses for Protocols (PRISMA-P) [24,25] checklist was used to ensure appropriate reporting of the protocol at hand (Appendix A).

### 2.1. Inclusion Criteria

The population, concept, and context (PCC) mnemonic was used to define the review question, as recommended by the JBI methodology for scoping reviews [21,23]. Consequently, we also used the PCC mnemonic to define the inclusion criteria. Thereafter, the types of sources eligible for inclusion are clarified in detail.

#### 2.1.1. Participants

Studies involving living humans of all ages and sexes who underwent quantitative MRN of the peripheral nerves will be considered. Except for the exclusion of peripheral nerve tumors such as fibromas, schwannomas, metastases, and peripheral nerve sheath tumors, no criteria will be applied to the health condition. Peripheral nerve tumors will not considered be as they represent neoplastic neuropathies possibly containing non-neural tissue. The imaged peripheral nerves will include nerves originating from the spinal cord segments (cervical, thoracic, and lumbar), the plexuses and the nerves in the extremities. Studies involving cranial nerves will be excluded as the used methodologies differ considerably from the aforementioned peripheral nerves. Animal studies will be excluded.

#### 2.1.2. Concept

Literature that describes a methodology for quantitative MRN and/or reports measured values of MR parameters will be considered. MR parameters that are potentially quantified with MRN are: (i) T1 relaxation time, (ii) T2 relaxation time, (iii) proton density, (iv) magnetization transfer, (v) diffusion tensor imaging parameters, (vi) perfusion, (vii) susceptibility, (viii) morphometry, (ix) microstructure. Literature that describes methodologies and/or reports measured values of MR parameters that are not listed will also be considered, as long as they are quantitative. The magnetic field strengths considered will range from 1.5 to 7 tesla, as they are clinically the most relevant.

#### 2.1.3. Context

The scoping review will consider settings where the target participants undergo quantitative MRN. However, settings where MRN occurred after surgery, i.e., post-operative MRN, will be excluded as the surgical procedure is invasive and, therefore, reflects iatrogenic changes in the peripheral nerves. There will be no limitations regarding a particular healthcare setting or geographic location.

#### 2.1.4. Types of Sources

The scoping review will consider peer-reviewed journal and conference articles in the English language, as this is the primary source of information in this field of research. The study designs considered for inclusion will be experimental (randomized controlled trials, non-randomized controlled trials), quasi-experimental (before and after studies, interrupted time-series studies), and analytical observational (prospective and retrospective cohort studies, case-control studies, analytical cross-sectional studies). For example, this will include studies that compare a healthy cohort with a cohort of a specific neural disease, healthy cohorts of different ages, sex, or conditions (smoking/non-smoking), pre- with post-traumatic nerve injury, but also quantitative parameters of different peripheral nerves of the same subjects. Descriptive studies of all types (e.g., cross-sectional studies, case series, individual case reports) will be excluded due to the lack of a comparison group. Systematic reviews that meet the inclusion criteria will also be considered. Studies describing methodologies will be included without any restriction on study design. Unpublished and gray literature, conference abstracts, and non-systematic reviews will be excluded to avoid duplication of data and to ensure the feasibility of the review.

### 2.2. Search Strategy

The search strategy will aim to identify all eligible studies. To initially identify studies on the topic, a limited search of MEDLINE (PubMed) was undertaken. A full search strategy was then developed for MEDLINE (PubMed), EMBASE (Ovid), Web of Science, and Scopus. The search terms of the full search strategy base on the text words contained in the titles and abstracts and the index terms used to describe relevant studies from the limited search. The search queries for all four databases are available online at https://github.com/fabianbalsiger/qmrn-review (accessed on 28 April 2022). For MEDLINE (PubMed), the search query is also available in Appendix A. We adapted the search strategy for all databases individually to the particularities of the used platform to access the database. We will search the databases again before the final analyses to identify any recent studies meeting the inclusion criteria. All searches will be limited to the English language. No limitations regarding the date of publication will be imposed. We will screen the reference lists of included studies or relevant reviews for additional studies. In case of missing data, the authors of primary studies will be contacted.

### 2.3. Study Selection

All identified sources by the full search will be uploaded into Covidence (Veritas Health Innovation, Melbourne, Australia), which will automatically remove duplicates. Two or more independent reviews will then screen the title and abstract of each source against the inclusion criteria. To enable full-text screening, the full text of potentially relevant sources will then be retrieved and uploaded to Covidence. The full text of each source will then be assessed by two or more independent reviewers against the inclusion criteria. The reasons for the exclusion of sources during the full-text screening will be reported in the scoping review. At each stage of the study selection, disagreements between the reviewers will be resolved through discussion or with an additional reviewer/s. In a PRISMA-ScR flow diagram [20], the results of the search and the study selection will be reported. A pilot test will be conducted with the reviewers at each stage to ensure consistency in the study selection.

### 2.4. Data Extraction

The data will be extracted from included studies by two or more independent reviewers. The reviewers will develop a tailored tool for the data extraction. Key information to be extracted will include the health condition, sample size, demographic parameters (age and sex), type of peripheral nerve imaged, MR scanner vendor, type, and field strength, MR sequence type and parameters, image reconstruction and processing setting/parameters, values and ranges of the measured MR parameters. The data extraction will differ depending on the type of MR parameter investigated (e.g., the b-value will be extracted for diffusion tensor imaging but will not be available for T2 relaxometry). A draft of the data extraction is provided in Table 1. If necessary, the draft of the data extraction will be modified and revised during the data extraction. We will detail any modifications in the scoping review. Disagreements between the reviewers in the extracted data will be resolved through discussion, or with an additional reviewer/s. Missing or additional data will be requested from authors of studies where required.

### 2.5. Data Analysis and Presentation

A basic descriptive analysis will be conducted (e.g., count of studies per MR parameter, imaged peripheral nerves) to get a sense of the current state of research in quantitative MRN. We will present the extracted data in diagrammatic or tabular form, categorized by the measured MR parameter. To relate the extracted data to the review objective and questions, a narrative summary will be provided. Based on this summary, guidance on how to perform quantitative MRN nowadays will be given.

## 3. Discussion

The aim of the scoping review will be to consolidate the literature on quantitative MRN of human peripheral nerves. The characteristics and ranges of the methodologies as well as the measured values of MR parameters will be extracted from the literature and reported appropriately.

The scoping review has a broad review question aiming to map the currently available literature on quantitative MRN. Therefore, no restrictions on the type of MR parameter will be imposed on the literature search. Most importantly, not only the measured values of MR parameters but also the characteristics of the used methodologies to measure the MR parameters will be extracted and reported in the scoping review. This will map the extent of the literature on quantitative MRN regarding both clinical and technical aspects.

A considerable amount of literature is expected to be eligible for full-text screening and also for inclusion in the scoping review due to the broad review question. To ensure the feasibility of the scoping review, we opted to limit the study by certain exclusion criteria (e.g., neoplasms, cranial nerves, surgery, types of sources, English language) while retaining a broad inclusion regarding the review question. Hence, the inclusion criteria do not impose any limitations regarding methodology and measured MR parameters as long as it is quantitative MRN.

In conclusion, the scoping review will provide a detailed summary of the available literature on quantitative MRN. The findings will be helpful for all stakeholders in this field of research to get an overview, identify gaps in the literature, and identify possible future work.

## Figures and Tables

**Table 1 mps-05-00039-t001:** Draft of the anticipated data to be extracted.

Data	Explanation
Reference	Author(s), year of publication, study title.
Cohorts/disease	Description of the cohorts/disease(s) examined in the study.
Sample size	Sample size per cohort (total / female / male).
Age	Age characteristics of the cohorts (e.g., mean ± standard deviation).
Peripheral nerve	The examined peripheral nerve(s).
MR scanner	Vendor and type of the MR scanner.
Field strength	The magnetic field strength in tesla.
Coil	Characteristics of the coil(s).
MR sequence type	The type of the MR sequence (e.g., diffusion tensor imaging, turbo spin echo).
MR sequence parameters	Parameters of the MR sequence (e.g., fat saturation, echo time, repetition time, voxel size, field of view, matrix size, acquisition time, anatomical plane).
Image processing	Information on any specific image reconstruction and processing (e.g., manual region of interest placement in specific software).
MR parameter type	The type of the measured MR parameter (e.g., T1/T2 relaxation time, magnetization transfer ratio, cross-sectional area).
MR parameter value	The value of the MR parameter per cohort and peripheral nerve (e.g., mean ± standard deviation).
Comments	Any information that does not fit in any of the other fields but will be deemed relevant.

## Data Availability

The search queries are available online at https://github.com/fabianbalsiger/qmrn-review (accessed on 28 April 2022).

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
