# Peer review of "Methodologies and MR Parameters in Quantitative Magnetic Resonance Neurography: A Scoping Review Protocol"

_mps, 2022, doi:10.3390/mps5030039_

Round 1

Reviewer 1 Report

The authors are to be congratulated on their work.

The protocol was thoroughly prepared according to the correct methodology. Some minor remarks:

  • In the introduction it is stated that only three SR were found that reported on MRN parameters. However more literature is available. We published a similar SR on cranial nerve MRN (https://pubmed.ncbi.nlm.nih.gov/32401614/).
  • The review question (paragraph 1.1) does not state if you will include healthy subjects or patients with some neural disease (traumatic, systemic,...), this will hugely affect the SR. Because now it is unclear if you are aiming to find MRN "benchmarking" parameters for the peripheral nerves or are you in search for MRN biomarkers that are disease specific? Please elaborate on this as it is pivotal for your SR.
  • 2.1.3 Context: you say you will exclude MRN in post-operative cases as these might reflect iatrogenic injuries but what about articles that look at post-traumatic nerve injuries, these can be considered iatrogenic as well. This also confirms my previous point: are you looking for benchmarking values of MRN parameters in healthy subjects or not? This is not clear to me. 
  • 2.4 Data extraction. You will include quantitative parameters which is of course necessary but have you thought about including more qualitative parameters such as presence of artifacts, observer agreement, ...

Thank you for considering these comments. 

Author Response

We thank the reviewer for the very helpful comments.

Point 1: In the introduction it is stated that only three SR were found that reported on MRN parameters. However more literature is available. We published a similar SR on cranial nerve MRN (https://pubmed.ncbi.nlm.nih.gov/32401614/).

Response 1: Thank you for pointing out the SR. We added it to the introduction accordingly.

Point 2: The review question (paragraph 1.1) does not state if you will include healthy subjects or patients with some neural disease (traumatic, systemic,...), this will hugely affect the SR. Because now it is unclear if you are aiming to find MRN "benchmarking" parameters for the peripheral nerves or are you in search for MRN biomarkers that are disease specific? Please elaborate on this as it is pivotal for your SR.

Response 2: The SR will include studies that perform a comparison between cohorts/groups, i.e., both MRN “benchmarking” and biomarkers. For instance, the cohorts could be healthy and neural disease but also healthy smokers and non-smokers. Further, it will also include studies that compare, for instance, healthy women and men or DTI parameters of different lumbar nerve roots. Therefore, we think our review question is adequate but we clarified Section 2.1.4. Types of sources with some examples.

Point 3: 2.1.3 Context: you say you will exclude MRN in post-operative cases as these might reflect iatrogenic injuries but what about articles that look at post-traumatic nerve injuries, these can be considered iatrogenic as well. This also confirms my previous point: are you looking for benchmarking values of MRN parameters in healthy subjects or not? This is not clear to me.

Response 3: Post-traumatic nerve injuries will be included. We don't consider them as iatrogenic, as the injury is not a result of medical activity (as surgery is). We included this example as part of the clarification of 2) in Section 2.1.4.

Point 4: 2.4 Data extraction. You will include quantitative parameters which is of course necessary but have you thought about including more qualitative parameters such as presence of artifacts, observer agreement, ...

Response 4: The main focus are clearly quantitative parameters. If we deem more qualitative parameters as relevant (e.g., a subjective analysis of artifacts in quantitative MRN), we will include it. The “Comments” field in the data extraction draft is intended for this purpose: “Any information that does not fit in any of the other fields but will be deemed relevant”. We also may modify the data extraction should we identify that we missed an important aspect, as is written in the manuscript “The draft will be modified and revised as necessary during the process of extracting data”.

Reviewer 2 Report

The authors present an extensive review on the magnetic resonance neurography technique, which is around since quite some time already, yet has been limited in its use to specialized centers, Heidelberg being the most prominent example. The use of "quantitative" MRN is an even newer field that potentially helps elevating this method further.

The extensive research that has been performed in recent years is incredible and the authors aim to bundle this in their review.

Of special merit here is that the authors aim to identify gaps in the literature and look into the future with follow-up research that is needed in this field. Clear recommendations or at least general guidelines/guidance on how to perform quantitative MRN together with most used sequences should be added as well in this review.

Author Response

We thank the reviewer for the general comments and the suggestion about providing guidance on how to perform quantitative MRN.  

Point 1: Of special merit here is that the authors aim to identify gaps in the literature and look into the future with follow-up research that is needed in this field. Clear recommendations or at least general guidelines/guidance on how to perform quantitative MRN together with most used sequences should be added as well in this review.

Response 1: We will include such a section in the scoping review. Therefore, we revised the methods section “2.5 Data analysis and presentation” by adding the sentence “Guidance on how to perform quantitative MRN nowadays will be given.”.

Reviewer 3 Report

His manuscript presents a protocol for a scoping review on MRN. 

the methodology is well described and seems in line with similar scoping reviews.

minor comments:

some sentences describe procedures in the past or present tense, while others use the present tense. Please ensure all tenses are correct and reflect a future action.

Author Response

We thank the reviewer for the comments. 

Point 1: some sentences describe procedures in the past or present tense, while others use the present tense. Please ensure all tenses are correct and reflect a future action.

Response 1: As suggested, we revised the English language of our manuscript accordingly. Please find the changes highlighted in blue.